# Speech-T: Transducer for Text to Speech and Beyond

**Jiawei Chen**
South China University of Technology
csjiaweichen@mail.scut.edu.cn

**Xu Tan**
Microsoft Research Asia
xuta@microsoft.com

**Yichong Leng**
University of Science and Technology of China
lyc123go@mail.ustc.edu.cn

**Jin Xu**
Tsinghua University
j-xu18@mails.tsinghua.edu.cn

**Guihua Wen**
South China University of Technology
crghwen@scut.edu.cn

**Tao Qin**
Microsoft Research Asia
taoqin@microsoft.com

**Tie-Yan Liu**
Microsoft Research Asia
tyliu@microsoft.com

## Abstract

Neural Transducer (e.g., RNN-T) has been widely used in automatic speech recognition (ASR) due to its capabilities of efficiently modeling monotonic alignments between input and output sequences and naturally supporting streaming inputs. Considering that monotonic alignments are also critical to text to speech (TTS) synthesis and streaming TTS is also an important application scenario, in this work, we explore the possibility of applying Transducer to TTS and more. However, it is challenging because it is difficult to trade off the emission (continuous mel-spectrogram prediction) probability and transition (ASR Transducer predicts blank token to indicate transition to next input) probability when calculating the output probability lattice in Transducer, and it is not easy to learn the alignments between text and speech through the output probability lattice. We propose SpeechTransducer (Speech-T for short), a Transformer based Transducer model that 1) uses a new forward algorithm to separate the transition prediction from the continuous mel-spectrogram prediction when calculating the output probability lattice, and uses a diagonal constraint in the probability lattice to help the alignment learning; 2) supports both full-sentence or streaming TTS by adjusting the look-ahead context; and 3) further supports both TTS and ASR together for the first time, which enjoys several advantages including fewer parameters as well as streaming synthesis and recognition in a single model. Experiments on LJSpeech datasets demonstrate that Speech-T 1) is more robust than the attention based autoregressive TTS model due to its inherent monotonic alignments between text and speech; 2) naturally supports streaming TTS with good voice quality; and 3) enjoys the benefit of joint modeling TTS and ASR in a single network.

## 1 Introduction

Transducer [7] is a sequence-to-sequence model widely used in automatic speech recognition (ASR) [19, 17, 36, 28]. ASR Transducer consists of a speech encoder that converts an input speech sequence to hidden representations, a text encoder that processes already generated text tokens autoregressively, and a joint network that predicts the next text token. Transducer networks are trained to maximize the alignment probability between speech and text sequences, with the help of a forward algorithm on the output probability lattice (see Figure 1 for more details). The alignments

35th Conference on Neural Information Processing Systems (NeurIPS 2021).

learned in the output probability lattice of the Transducer are strictly monotonic, thus well modeling monotonic alignments between speech and text sequences in ASR. By restricting the speech encoder only to see the previous speech frames, Transducer naturally supports streaming inputs and is the one of the most popular solutions for streaming ASR [36, 12, 33, 3].

Although Transducer is originally designed for ASR, clearly, its advantages of modeling monotonic alignments between inputs and outputs and supporting streaming inputs perfectly match the requirements of text to speech (TTS) synthesis [26, 30, 22, 1, 6, 16, 13, 20, 21, 11]. First, learning alignments between text and speech is important in TTS [26], and different approaches such as attention mechanisms [22, 8, 25, 13] or duration prediction [20, 21, 11, 4] have been leveraged to model the alignments. However, attention mechanisms suffer from unstable alignments with word skipping and repeating issues [16, 20, 8], and duration prediction suffers from additional complication because of the training of duration predictor and training-inference mismatch caused by the predictor [20, 21, 11]. Thus, the inherent monotonic alignments in Transducer can be a quite competitive solution for TTS over previous methods. Second, streaming TTS (or incremental TTS) [14, 5, 23], which aims to support streaming text inputs, can greatly save the synthesis latency and can be widely used in online scenarios such as conversations. Unfortunately, existing solutions for streaming TTS [14] mostly rely on the argmax operation of the encoder-decoder attention values to decide the end of speech frame corresponding to an input word, which is not accurate and robust. Transducer can naturally support streaming inputs and can be a better solution compared with previous methods for streaming TTS. Therefore, we explore the possibility of applying Transducer for TTS in this work.

However, applying Transducer to TTS is challenging due to the distinctive characteristics of speech (e.g., mel-spectrograms) generation in TTS compared with the text token generation in ASR. First, there are two actions in the output probability lattice of Transducer [7, 32]: emission that predicts a text token and transition that predicts a blank token to indicate null outputs in current step and the transition to the next input speech frame [7]. In ASR Transducer, the blank token is added into the token vocabulary, which means that the emission (token prediction) and transition (blank prediction) can be modeled in a unified probability distribution through the softmax on all the vocabulary tokens. However, mel-spectrograms in TTS are continuous, which makes it hard to trade off the two probabilities when calculating the output probability lattice in TTS Transducer. As shown in our experiments, ill tradeoff between the two probabilities causes the failure of alignment learning. Second, while predicting current mel-spectrogram, previous mel-spectrograms can provide enough information as they are very similar due to the continuity between consecutive mel-spectrograms [22, 2], which may lead to copying the previous frame instead of learning information from the text encoder and thus harms alignment learning. Moreover, Transducer learns the appropriate alignment path from huge candidate paths in the probability lattice, which further causes difficulties in alignment learning.

In this work, we propose a Transformer [29] based Transducer model, named SpeechTransducer (Speech-T for short), for TTS. First, we design a lazy forward algorithm that separates transition prediction from mel-spectrogram prediction when deriving the loss function of Transducer. Specifically, this lazy forward algorithm on the probability lattice only calculates the probability of transition/nontransition (using a binary classification), without the probability of mel-spectrogram prediction. Second, we introduce a diagonal constraint [25, 16, 2] in the probability lattice to assist alignment learning, which ignores the alignment paths that deviate too much from the diagonal. The intuitive idea behind this design is that the alignments between text and speech should be monotonic, and thus lie in the diagonal region in the probability lattice. Third, by adjusting the text encoder of Speech-T to look the whole context or only the previous context, it can support full-sentence TTS or streaming TTS. Last but not the least, we further extend Speech-T to support both TTS and ASR together for the first time, which enjoys several advantages including fewer parameters as well as streaming synthesis and recognition in a single model. Experiment results on LJSpeech datasets demonstrate that Speech-T 1) is more robust than attention based autoregressive TTS models due to its inherent monotonic alignments between text and speech, 2) can naturally support streaming TTS with good voice quality, and 3) enjoys the benefit of joint modeling TTS and ASR in a single network.

The main contributions of this work are summarized as follows:

- To leverage the advantages of Transducer (i.e., monotonic alignment modeling and streaming input supporting), we propose SpeechTransducer for TTS. To the best of our knowledge, we are the first to achieve competitive voice quality in TTS and support streaming TTS using Transducer.

- We design a lazy forward algorithm to trade off the two probabilities of mel-spectrogram prediction and transition prediction, and design a diagonal constraint to ensure the alignment learning between text and speech, to ensure the quality of SpeechTransducer.

- We further extend SpeechTransducer to support both TTS and ASR at the same time, which enjoys several advantages including compact modeling (fewer parameters), streaming synthesis and recognition, self-ranking, etc. To the best of our knowledge, we are the first to successfully perform the two tasks with a single model.

## 2   Background

In this section, we first introduce the basic formulation of neural Transducer, and then describe the application of Transducer on several sequence-to-sequence tasks.

**Formulation of Transducer**   Neural Transducer is originally proposed to model the sequence transduction between input speech and output text sequences in ASR [7]. It learns strictly monotonic alignments between speech (e.g., mel-spectrogram) sequence $X = \{x_1, x_2, ..., x_U\}$ and output text (e.g., phoneme or character) sequence $Y = \{y_1, y_2, ..., y_T\}$, where $U$ and $T$ are the length of speech and text sequences respectively (U is short for utterance, T is short for text). A Transducer contains 1) a speech encoder which extracts acoustic representations from the speech sequence $X$, 2) a text encoder works autoregressively to extract hidden representations from the text sequence $Y$, and 3) a joint network to predict the next token in $Y$ based on the combination between the output hidden representations of the speech encoder and text encoder.

This combination is similar to Cartesian product, where each hidden representation of speech is concatenated with each hidden representation of text to get a output probability lattice with a shape of $(U, T)$, as shown in Figure 1. Each node $(m, n)$ in the output probability lattice represents that speech sequence $X_{1:m}$ is aligned to text sequence $Y_{1:n}$. Each vertical edge in the lattice represents an emission (i.e., a normal token prediction) and the probability of this edge is the corresponding probability of token prediction. Each horizontal edge represents a transition to the next input element (i.e., a blank token is added into the output vocabulary to represents null output) and the probability of this edge is the corresponding probability of blank token prediction. Each path from the bottom left to top right in the lattice represents a possible alignment between the speech and text sequence.

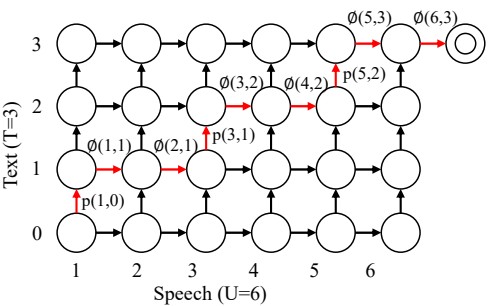

Figure 1: The output probability lattice of ASR Transducer. The red path is a possible alignment between the speech and text sequences.

During the training of Transducer, it marginalizes all the alignments (paths) $\mathcal{A}$ between $X$ and $Y$ over the probability lattice shown in Figure 1 and minimizes the negative log-probability of a conditional distribution:

$$\mathcal{L}_{\text{asr}} = -\log P(Y|X) = -\log \sum_{\alpha \in \mathcal{F}^{-1}(\mathbf{y})} P(\alpha|X), \tag{1}$$

where $\mathcal{F}^{-1}$ is the inverse function of $\mathcal{F}$ that removes the blank token $\phi$ from each possible alignment $\alpha$ between $X$ and $Y$. Each $\alpha$ consists of $U$ blanks ($\phi$) and $T$ tokens $(y_1, y_2, y_3, \ldots, y_T)$. For example, we refer $\alpha = (y_1, y_2, \phi, \phi, y_3, \phi, \phi)$ as a possible alignment between $X = (x_1, x_2, x_3, x_4)$ and $Y = (y_1, y_2, y_3)$, then $\mathcal{F}(\alpha)$ becomes $Y$.

Computing Equation (1) by summing all possible alignments naively is computationally intractable. Therefore, Graves [7] introduced an efficient forward algorithm to handle this problem. In brief, it defines a forward variable $\alpha(u, t)$ to represent the probability of outputting $Y_{1:t}$ given $X_{1:u}$. $\alpha(u, t)$ can be obtained by recursive calculation:

$$\alpha(u, t) = \alpha(u - 1, t)\phi(u - 1, t) + \alpha(u, t - 1)p(u, t - 1), \tag{2}$$

where $\phi(u - 1, t)$ represents the transition probability (predicting blank token given $x_{u-1}$ and $y_t$), $p(u, t - 1)$ represents the emission probability (predicting token $y_t$ given $x_u$ and $y_{t-1}$), and the initial condition is $\alpha(1, 0) = 1$.

**Application of Transducer** Neural Transducer network has several advantages: 1) Monotonic alignment modeling. The alignment path in the output probability lattice is inherent monotonic from bottom left to top right, which makes it suitable for the tasks with monotonic alignments between the input and output sequences, e.g., ASR. 2) Streaming input support. Transducer can generate text based on streaming speech input, by making the speech encoder only to see the previous speech context and using the transition (blank token) to determine the finish of text generation based on current input speech segment. Due to these advantages, Transducer has been widely used in ASR for streaming scenarios [19, 17, 9]. RNN was used as the backbone of speech encoder and text encoder in early ASR Transducer (i.e., RNN-T) [7], and then Transformer [29] was introduced into Transducer (i.e., Transformer-T [36, 28]) to enhance the ability of modeling long-term dependency. Besides ASR, Transducer can also be applied on other sequence-to-sequence tasks such as text summarization [34, 35], morphological inflection [34], and machine translation [35].

The application of Transducer on TTS is still under-exploited. Yasuda et al. [32] leveraged the monotonic alignment of Transducer in TTS, but generated speech with poor voice quality. The reasons include: 1) It is hard to trade off the probability of transition and emission (mel-spectrogram prediction) when calculating the output probability lattice, since mel-spectrogram and transition predictions are modeled separately (mel-spectrogram prediction using regression, while transition prediction using binary classification). 2) The alignments between speech and text in TTS are more difficult to learn than those in ASR. The voice generated by Yasuda et al. [32] usually has incorrect alignments such as extremely long duration for a certain phoneme. In our work, we apply Transducer to TTS by reformulating the forward algorithm with a lazy probability lattice to avoid the troublesome probability tradeoff, and adding a diagonal alignment constraint on the loss to facilitate the alignment learning. We also adjust the Transudcer model to support streaming TTS and further support both TTS and ASR in a single Transducer model, which can bring many interesting applications to explore.

## 3 SpeechTransducer

In this section, we first introduce the overall model architecture of our proposed SpeechTransducer (Speech-T for short) in Section 3.1. Then we introduce our proposed lazy forward algorithm on the probability lattice in Section 3.2. Next, we propose a diagonal constraint on the alignment paths to help the alignment learning in Section 3.3. Based on the proposed lazy forward algorithm and alignment constraint, we derive our loss function of Speech-T in Section 3.4. Moreover, we describe the design of Speech-T for streaming TTS in Section 3.5. At last, we extend Speech-T to support both TTS and ASR in a single model in Section 3.6.

### 3.1 Overall Model Architecture

As shown in Figure 2(a), Speech-T consists of a text encoder that converts an input text (e.g., phoneme) sequence to hidden representations, a speech encoder that processes already generated speech (e.g., mel-spectrogram) frames autoregressively, and a joint network that processes the combination of text and speech hidden representations to predict the next speech frame. Each token in text sequence $X$ with length of $T$ is converted into embedding while the speech sequence $Y$ with length of $U$[1] is processed by a Pre-Net with several convolutional layers, and then taken as input to the text and speech encoder respectively. As shown in Figure 2(b) and 2(c), both the text and speech encoders are based on Transformer [29], which consists of several blocks where each block contains a self-attention network and a 1D convolutional network [20], both followed by a residual connection and a layer normalization. A difference between the two encoders is that the speech encoder adopts causal self-attention and convolution to only see the previous context to support the autoregressive prediction in joint network. As shown in Figure 2(d), the text and speech hidden representations with length of $T$ and $U$ respectively are processed by a linear layer and then concatenated together in a way of Cartesian product, resulting in a shape of $(T, U)$, which are further processed by several linear layers to get the output probability lattice to predict speech frames during training. We describe the specific designs to train Speech-T in the following subsections.

---

[1]In both TTS and ASR, we use $T$ and $U$ to represent the length of text and speech sequence respectively, and use $X$ and $Y$ to represent the input and output sequence respectively.

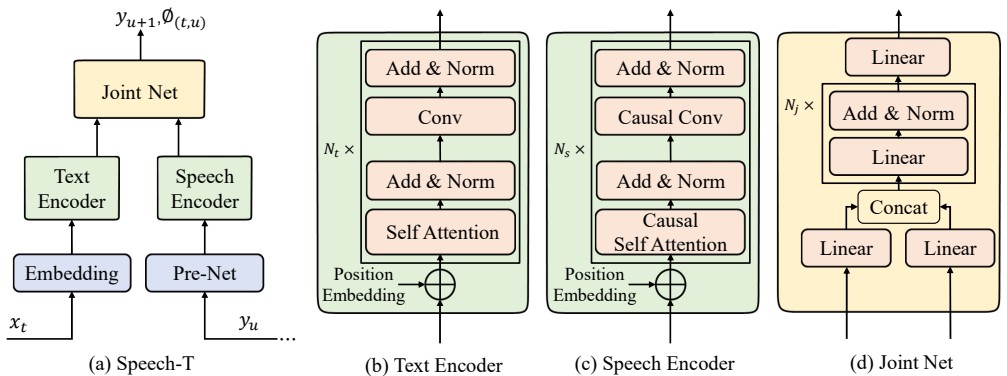

Figure 2: (a) The overall model structure of Speech-T. (b) Text Encoder. (c) Speech Encoder. (d) Joint Net, whose output dimension is 81, where the first 80 dimension is for mel-spectrogram prediction and the remaining 1 dimension is for transition prediction.

## 3.2 Lazy Forward Algorithm

In ASR Transducer (e.g., RNN-T), it is easy to trade off the probability of emission (text token prediction) and transition (blank token prediction) when calculating the Transducer loss since the blank token is added into the token vocabulary and all tokens are modeled in a unified probability distribution through a single softmax operation. However, mel-spectrograms in TTS are continuous and modeled as a regression problem while transition prediction is modeled as a classification problem, which makes it hard to trade off the two probabilities in TTS Transducer. Thus, we design a lazy forward algorithm on the probability lattice to separate transition prediction from mel-spectrogram prediction when deriving the loss function of Transducer. Specifically, we only calculate the probability of transition/non-transition (by using a binary classification) to get the forward variable $\alpha(t, u)$, without taking the probability of mel-spectrogram prediction into account:

$$\alpha(t, u) = \alpha(t - 1, u)\phi(t - 1, u) + \alpha(t, u - 1)(1 - \phi(t, u - 1)), \tag{3}$$

where $\phi(t - 1, u)$ is the probability of transition at node $(t - 1, u)$ (i.e., the combination of the hidden representations corresponding to text token $x_{t-1}$ and speech frame $y_u$). The forward algorithm is called *lazy* because we do not eagerly calculate the exact emission probability in Equation (3). Instead, we use a probability $1 - \phi(t, u - 1)$ to represent the overall probability of emission, but do not care about the exact probability of the specific mel-spectrogram prediction.

There are some previous works [18] also modeling transition/non-transition as a binary classification in a similar way (Equation 13 and 14 in Raffel et al. [18]). However, there are several differences: 1) The formulation in Raffel et al. [18] is used to ensure the encoder-decoder attention (e.g., in automatic speech recognition, text summarization, and neural machine translation) is monotonic, and does not handle the specific/detailed emission probabilities, i.e., the emission probability of each target token like we do; 2) Our scenario is different from theirs and they can not address the problems we handle in text to speech. We also noticed a blog introducing similar formulation during paper rebuttal. However, there are several differences: 1) The motivation of proposing our lazy forward algorithm is that we found it is hard to trade off the two probabilities if using traditional Transducer formulation in our preliminary experiments. Although we tried different weights to balance the two loss terms, we fail to find a stable one, which motivates us to propose some alternative methods; 2) We model the transition and emission prediction in a single joint network but not separate networks, and we design alignment constraint on the lazy forward algorithm to derive the final loss function, since we found it is hard to converge if not using any constraint on the calculation of the probability lattice, and we also formulate our Transducer in a streaming way and extend it to model unified TTS/ASR in a single Transducer model; 3) We conduct comprehensive experimental studies to verify the effectiveness of the proposed method, but not a pure brainstorming formulation.

## 3.3 Alignment Constraint

Ideally, the transition prediction can be learned well through the lazy forward algorithm and thus good alignments can be obtained through the learned transition. However, it is challenging for the

alignment learning (transition prediction) in TTS, due to several reasons: 1) While predicting current mel-spectrogram, previous mel-spectrograms can provide enough information as they are very similar due to the continuity between consecutive mel-spectrograms, which leads to copying the previous frame instead of learning information from the text encoder and thus harms alignment learning. Taking the lattice in Figure 3 as an example, due to copy, the learned alignment path may first go up to reach the top and then go right to reach the end, or first go right to reach the rightmost and then go up to reach the end, both of which are unreasonable and deviate too much from the diagonal. 2) Transducer is trained to learn the appropriate alignment path by marginalizing the huge candidate paths over the probability lattice and thus causes difficulties in alignment learning.

Based on prior knowledge, the alignment path should be monotonic and lie in the diagonal region in the probability lattice, which could be a good inductive bias to help the alignment learning. Thus, we introduce a diagonal constraint in the probability lattice to assist alignment learning, as shown in Figure 3, which ignores the alignment paths that deviate too much from the diagonal. Our alignment constraint works as following steps: (1) We construct an alignment path, as shown in the red path in Figure 3, by aligning the text and speech sequence in training data with some alignment tools (e.g., MFA [15]). (2) To ensure the flexibility in the constraint, we extend the alignment path by $\tau$ frames in both the left and right along the horizontal axis to form a diagonal and banded region, as shown in the green area in Figure 3. (3) We only take the path within this diagonal and banded region into account for loss calculation, and thus our Speech-T can learn the alignments (transition prediction) better.

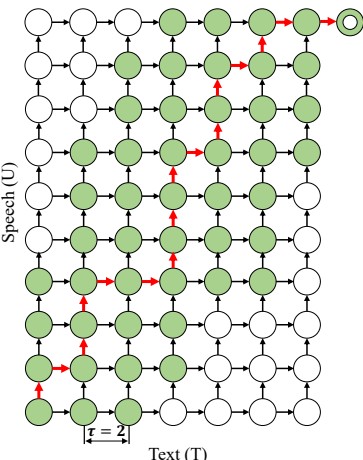

Figure 3: The diagonal and banded region for alignment constraint. We take $\tau = 2$ as an example.

## 3.4 Loss Function

Based on the proposed lazy forward algorithm and alignment constraint, we can derive the loss function of our proposed Speech-T for TTS as

$$\mathcal{L}_{\text{tts}} = \sum_t \sum_u \mathbb{I}\{(t, u) \in \text{band}\}\alpha(t, u)(1 - \phi(t, u))|y_{u+1} - f(t, u)|, \tag{4}$$

where $\mathbb{I}\{(t, u) \in \text{band}\}$ is an indicator function to indicate whether the index $(t, u)$ is in the constraint banded region of the probability lattice or not. We will only calculate the loss from the indexes that are in the banded region. $\alpha(t, u)$ is the forward variable calculated in Equation (3), and $1 - \phi(t, u)$ is the overall probability of emission at node $(t, u)$. The L1 loss $|y_{u+1} - f(t, u)|$ (which is calculated between the ground-truth mel-spectrogram frame $y_{u+1}$ and the predicted mel-spectrogram $f(t, u)$) is weighted by $\alpha(t, u)(1 - \phi(t, u))$, where the weight means the probability of emission at node $(t, u)$ by taking the forward variable into account. In this way, we do not need to trade off the probability of transition and a certain mel-spectrogram prediction. In our implementation, the joint network outputs a vector with 81 dimension, where the first 80 dimension is for mel-spectrogram prediction and the last dimension is followed by a sigmoid function to predict the transition.

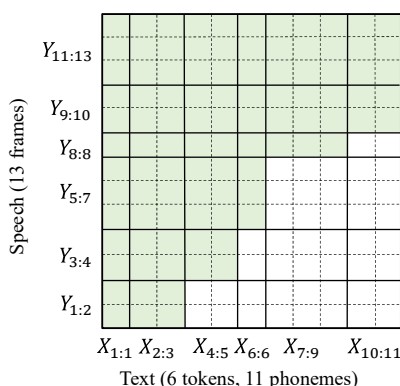

Figure 4: $k$ look-ahead mechanism, where we take $k = 1$ as an example. The green part means that the speech frames can see the context of the corresponding text tokens.

## 3.5 Streaming TTS with Speech-T

Due to the advantage of Transducer architecture used in Speech-T, it can naturally support streaming TTS [14, 5, 23]. When a text token arrives, Speech-T predicts mel-spectrogram autoregressively, until a transition is predicted, which indicates the end of generation for this text token. After that, the Speech-T will wait for the next text token in the

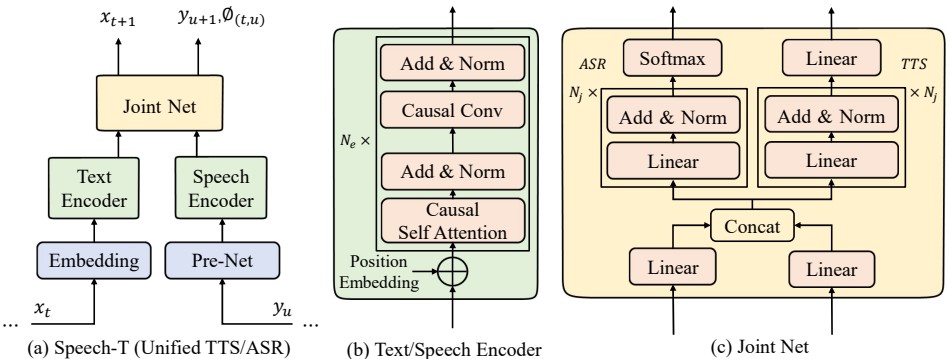

Figure 5: (a) The overall model structure of Speech-T for unified TTS/ASR. (b) Text/Speech Encoder. (c) Joint Net, where the output dimension is 81 (same with that in Figure 2) for TTS, and vocabulary size plus 1 (blank token) for ASR.

streaming inputs. We restrict each token in the encoder to only see the previous tokens. Alternatively, to trade off the latency and accuracy in streaming TTS, we can allow the token in the encoder to see $k$ future tokens ($k$ look-ahead, where $k = 0$ degenerates to only see previous tokens) to extract more context information. In practice, a word that contains multiple phoneme tokens is coming as a whole during streaming inputs. Thus, we allow each phoneme token can see all the phoneme tokens in the current word, and use $k$ to represent the number of words instead of phonemes. We illustrate the $k$ look-ahead mechanism in Figure 4.

## 3.6 Unified TTS/ASR with Speech-T

Considering that the text and speech encoder in Transducer can serve for both TTS and ASR, in this section, we further extend our proposed Speech-T as a unified model to support both TTS and ASR at the same time. We illustrate the model architecture of Speech-T for unified TTS/ASR in Figure 5. TTS and ASR share the same text encoder, speech encoder, and also joint network, except that there are some task specific branchs in the joint network for TTS and ASR individually, as shown in Figure 5(c). The model structure of the text/speech encoder is shown in Figure 5(b), both restricting the current token/frame to only see the previous tokens/frames to support the autoregressive prediction of both speech and text. In this way, both TTS and ASR can support streaming inputs. The unified TTS/ASR based on Speech-T enjoys several advantages, including compact modeling (with fewer parameters), streaming synthesis and recognition, self-ranking and speech chain [27] in a single model. We demonstrate some preliminary experiment results in this paper and leave more studies on these advantages as future work.

# 4 Experiments and Results

In this section, we first introduce the experimental setup, and then show the results of our Speech-T for TTS in terms of audio quality and robustness, and further conduct some studies to evaluate the effectiveness of the designs in Speech-T. At last, we show the experiment results on unified TTS/ASR with Speech-T and discuss its potential benefits.

## 4.1 Experimental Setup

**Dataset** We conduct experiments on LJSpeech [10], a public speech dataset consisting of 13,100 English audio clips and corresponding text transcripts. The total length of the audio is approximately 24 hours. We randomly split the dataset into three parts: 12500 samples for training, 300 samples for validation and 300 samples for test. We use 22.05kHz sampling rate, extract the 80-dimensional mel-spectrogram with 50 ms window size and 12.5 ms hop size, and normalize mel-spectrograms to have zero mean and unit variance. We convert the text sequence into phoneme sequence with grapheme-to-phoneme conversion [24].

**Implementation Details**   We use TransformerTTS [13] as the baseline system, which is an autoregressive encoder-attention-decoder model using Transformer [29] as the basic model structure. Both the encoder and decoder consist of 6 Transformer blocks, each with a self-attention network and feed-forward network with convolution [20]. The number of self-attention head is set to 2, the dimension of embedding size and hidden size are both 256, and the inner dimension of feed-forward network is 1024. We adopt the encoder in TransformerTTS as our speech encoder and text encoder in Speech-T. According to the requirements in full-sentence TTS and steaming TTS, these encoders can utilize causal or non-causal self-attention and convolution. We train our model on 8 Tesla V100 GPUs with a batch size of 6 sentences on each GPU. We use the Adam optimizer with $\beta_1 = 0.9$, $\beta_2 = 0.98$, $\epsilon = 10^{-9}$ and follow the same learning rate schedule in [29]. It takes about 3 days for training. The predicted mel-spectrograms are transformed into audio waveform using a well-trained Parallel WaveGAN vocoder [31]. More details of the model configurations and training details are presented in supplemental material (Section A.1).

### 4.2   Results of Speech-T

**Audio Quality**   We first evaluate the perceptual quality of Speech-T by mean opinion score (MOS) test, where 20 native English speakers are involved to give quality score for the synthesized speech. We compare Speech-T with other systems: 1) Recording, the ground-truth speech; 2) GT+Vocoder, where the recording is converted to mel-spectrograms first and then converted back to speech by vocoder; 3) TransformerTTS [13], which is a Transformer based autoregressive TTS model with high voice quality.   All the systems (including

Table 1: The MOS results with 95% confidence intervals.

| Method | MOS |
| --- | --- |
| Recording | $4.00 \pm 0.07$ |
| GT+Vocoder | $3.85 \pm 0.08$ |
| TransformerTTS [13] | $3.77 \pm 0.07$ |
| Speech-T | $3.74 \pm 0.07$ |

Speech-T) use Parallel WaveGAN [31] as the vocoder for fair comparison. The MOS results are shown in Table 1. It can be seen that our proposed Speech-T can achieve similar voice quality with TransformerTTS, demonstrating that we can successfully introduce Transducer into TTS with high voice quality[2].

Table 2: The robustness results of TransformerTTS and Speech-T. The number is calculated in sentence level (e.g., a sentence that has multiple skipping errors will be counted only once). # Errors means the number of sentences with either or both repeating and skipping errors.

| Method | # Repeat | # Skip | # Errors | Error Rate |
| --- | --- | --- | --- | --- |
| TransformerTTS | 6 | 15 | 16 | 32% |
| Speech-T | 0 | 0 | 0 | 0 |

**Robustness**   Since Speech-T can learn monotonic alignments between text and speech, which can generate more robust speech compared with the TTS models (e.g., TransformerTTS) using attention mechanism for alignment learning [30, 13]. We conduct the robust test between Speech-T and TransformerTTS using 50 particularly hard sentences [20] to measure the ratio of word skipping and repeating errors[3]. The results are shown in Table 2. It can be seen that TransformerTTS, based on attention mechanism for alignment learning, has a much higher error ratio. Benefiting from the monotonic alignment in Transducer, Speech-T can generate all sentences with no repeating and skipping errors, achieving highly robust speech synthesis.

Table 3: The MOS results of streaming TTS with 95% confidence intervals.

| Method | MOS |
| --- | --- |
| Recording | $4.02 \pm 0.12$ |
| Speech-T ($k = \infty$) | $3.76 \pm 0.11$ |
| Speech-T ($k = 2$) | $3.68 \pm 0.13$ |
| Speech-T ($k = 1$) | $3.44 \pm 0.15$ |
| Speech-T ($k = 0$) | $3.14 \pm 0.19$ |

---

[2]The audio samples generated by Speech-T and the baseline systems can be found in `https://speechresearch.github.io/speechtransducer/`.

[3]These sentences include spellings, repeated numbers, single letters, and long sentences. We attach the 50 sentences in the supplemental material. (Section A.4)

**Streaming TTS**   We evaluate our Speech-T for streaming TTS (designed in Section 3.5). We compare the voice quality in the $k$ look-ahead mechanism with different $k$. The results are shown in Table 3. It can be seen that the voice quality increases with the increase of $k$, at a cost of increased latency. $k = \infty$ achieves the best quality, which is the full-sentence non-streaming TTS as we study in Section 4.2. The voice quality of the streaming Speech-T ($k = 2$) nearly catches up with that of the non-streaming Speech-T ($k = \infty$) .

## 4.3   Analysis of Speech-T

In this subsection, we conduct some analyses to verify the effectiveness of the designs in Speech-T, including the lazy forward algorithm and the alignment constraint.

**Lazy Forward Algorithm**   We compare our lazy forward algorithm with two baselines: 1) the default forward algorithm in Transducer ASR (similar to Equation 2 with swapped $u$ and $t$), where the emission probability $p$ is obtained from the mel-spectrogram $|y_u - f(t, u-1)|$ loss. 2) the forward algorithm in Yasuda et al. [32], which uses a Gaussian mixture model (GMM) to calculate the probability of mel-spectrogram prediction and uses a modified forward algorithm that still cannot address the probability tradeoff issue.

Table 4: The CMOS results of different forward algorithms.

| Method | CMOS |
|---|---|
| Lazy algorithm in Speech-T | 0 |
| Default algorithm [7] | N/A |
| GMM based [32] | -1.54 |

Note that except for the forward algorithm, all settings are the same to ensure fair comparison. The CMOS results are shown in Table 4. We can have several observations: 1) the default forward algorithm in Transducer ASR cannot generate any reasonable speech (we denote the CMOS as N/A), due to the intractable tradeoff between the probability of transition and emission in TTS. Our lazy algorithm can avoid the tradeoff issue and generate voice with good quality. 2) The voice quality by our lazy algorithm is much higher than that by GMM based [32], demonstrating the effectiveness of our algorithm.

**Alignment Constraint**   We further study the effectiveness of alignment constraint in Speech-T. We explore two different settings: 1) removing our proposed alignment constraint in Speech-T; 2) replacing our alignment constraint with a handcrafted alignment constraint, which only differs with our alignment constraint in step (1) as described in the second paragraph of Section 3.3. Instead of obtaining accurate alignment from external tool, in the handcrafted constraint, we first calculate the ratio between the number of mel-spectrogram frames $U$ and number of phoneme tokens $T$ for every text-speech pair, and use $U/T$ (after rounding) as the duration of each phoneme to get the alignment.

The CMOS results are shown in Table 5. It can be seen that 1) removing the alignment constraint in Speech-T cannot generate any reasonable speech (we denote the CMOS as N/A), and 2) replacing the alignment constraint with handcrafted constraint can make the voice quality much worse, demonstrating the effectiveness of our alignment constraint in Speech-T.

Table 5: The CMOS results on alignment constraint.

| Method | CMOS |
|---|---|
| Speech-T | 0 |
| Speech-T w/o alignment constraint | N/A |
| Speech-T with handcrafted diagonal constraint | -1.33 |

## 4.4   Results of Unified TTS/ASR

We conduct experiments to evaluate our extended Speech-T for unified TTS/ASR. The experimental setup follows that in Section 4.1. We compare the performance of both TTS and ASR between our unified Speech-T and the separate TTS and ASR models. Since our Speech-T for unified TTS/ASR works in streaming manner, the separate models also work in streaming manner (using causal self-attention and convolution both in

Table 6: Comparison between unified ASR/TTS model and separate TTS and ASR models.

| Method | CMOS | PER |
|---|---|---|
| Speech-T (Unified TTS/ASR) | 0 | 9.0% |
| Speech-T (TTS only) | -0.09 | / |
| Speech-T (ASR only) | / | 9.7% |

the text and speech encoder) for comparison. We measure phoneme error rate (PER) for ASR and MOS for TTS, and show the results in Table 6. It can be seen that the unified TTS/ASR achieves slightly better CMOS and PER compared with the separate streaming models, showing that Speech-T can not only enjoy compact model parameter by unifying TTS and ASR, but also boost the performance by joint training in a single model. We discuss other benefits of unified TTS/ASR with Speech-T in supplemental material (Section A.2) due to space limitation.

## 5   Conclusions and Future Work

Motivated by the advantages of Transducer such as efficiently modeling monotonic alignments between input and output sequences and naturally supporting streaming inputs, in this work, we proposed SpeechTransducer, with several designs to address the challenges when applying Transducer to TTS. Experiment results demonstrate that SpeechTransducer achieves much better robustness due to the advantages of monotonic alignments while enjoying good voice quality. The extension of SpeechTransducer to unified TTS/ASR demonstrates the flexibility of Transducer based architecture and its potential for many applications, such as streaming TTS/ASR, self-reranking, and TTS/ASR co-adaption for target speakers in a single model. A limitation of SpeechTransducer is that it suffers from slow inference speed compared with non-autoregressive TTS models. Thus, designing fast SpeechTransducer models is an interesting direction. Applying SpeechTransducer to more speech tasks such as voice conversion and speech enhancement will also be interesting areas to explore. We hope that our SpeechTransducer can inspire the community to explore more possibilities.

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
