# A Appendix

## A.1 Model Hyperparameters and Training Details

**Model Hyperparameters** Our Speech-T consists of a text encoder, a speech encoder and a joint network. The text encoder includes 6 Transformer blocks [13], each block has a self-attention network and a feed-forward network with convolution [20]. The structure of the speech encoder is similar to that of the text encoder, except that causal self-attention and convolution are adopted to only see the previous context to support the autoregressive prediction. The mel-spectrogram is first processed by a Pre-Net [13] that consists of 3 fully connected layers with ReLU activation, and then fed into the speech encoder. In the joint network, the hidden representations of the two encoders are first processed with a linear layer and then concatenated together in a way of Cartesian product. The concatenation is passed through 3 feed-forward network block, each block consists of two linear layers with input/output size of 256/1024 for the first layer and 1024/256 in the second layer, after which residual connection and layer normalization are added. We list hyperparameters and configurations of all models used in our experiments in Table 7.

Table 7: Hyperparameters of TransformerTTS and Speech-T.

| Module | Hyperparameter | TransformerTTS | Speech-T |
|---|---|---|---|
| Phone Embedding | Dimension | 256 | 256 |
| Pre-Net | Layers | 3 | 3 |
| | Hidden | 64 | 64 |
| | Dropout | 0.5 | 0.5 |
| Text/Speech Encoder | Layers | 6 | 6 |
| | Hidden | 256 | 256 |
| | Conv1D Kernel | 9 | 9 |
| | Conv1D Filter Size | 1024 | 1024 |
| | Attention Heads | 2 | 2 |
| | Dropout | 0.1 | 0.1 |
| Joint Net | Layers | / | 3 |
| | Hidden | / | 256 |
| | First Linear | / | 256/1024 |
| | Second Linear | / | 1024/256 |
| | Dropout | / | 0.1 |

**Training and Inference** We trained our model on 8 Tesla V100 GPUs with a batch size of 6 sentences on each GPU. In our experiments, we found that large batch size will result in better voice quality, so we update the gradient of network parameters with gradient accumulation of 5 times. The alignment constraint weight $\tau$ mentioned in Section 3.3 is set to 10. We use the Adam optimizer with $\beta_1 = 0.9$, $\beta_2 = 0.98$, $\epsilon = 10^{-9}$ and follow the same learning rate schedule in [29], and the training time is nearly three days. In the inference process, we set a threshold of 0.5 for the probability of transition to the next mel-spectrogram, as mentioned in Section 3.2, and the output mel-spectrograms of our Speech-T are transformed into audio samples using well-trained Parallel WaveGAN vocoder [31].

## A.2 Unified TTS/ASR with Speech-T

We extend our proposed Speech-T as a unified model to support both TTS and ASR by using our lazy forward algorithm for TTS and standard Transducer loss [7] for ASR. The unified TTS/ASR with Speech-T enjoys several advantages, including compact modeling (with fewer parameters), streaming synthesis and recognition, re-ranking and back transformation on low-resource training data.

Here we conduct experiments to verify the effectiveness of re-ranking and leave the explorations on other advantages as future work. After the unified TTS/ASR model is well-trained, it can employ reranking: using TTS to rerank the candidates generated by ASR, and using ASR to rerank the candidates generated by TTS. Specifically, when using ASR to rerank TTS candidates, we can first

get multiple candidates by inferring TTS model multiple times (dropout in the Per-Net can introduce randomness), and use ASR in the unified model to score these candidates (e.g., ASR likelihood). The ASR likelihood is used to rerank the candidates to get the final target sequence, which may lead to better voice quality. We simply show the results of using ASR to rerank TTS candidates in Table 8. It can been seen that the voice quality of TTS can be improved with re-ranking.

## A.3 Evaluation Method

We conducted Mean Opinion Score (MOS) test to evaluate the performance of our model. The MOS is a digital measure of speech quality judged by human beings, which is usually calculated based on a human rating service similar to Amazon Mechanical Turk. Each generated sample is rated by several raters on a scale from 1 (bad) to 5 (excellent) with 0.5 point increments, as shown in Table 9. In our test, each rater is required to wear headphone and be native English speaker, and then 50 samples were selected for blind evaluation and scoring. After collecting all the evaluations, the MOS score $\mu$ is estimated by averaging the scores $m_k$ from different testers $k$. In addition, we also calculated the 95% confidence intervals ($CIs$) for the score.

Table 8: Unified TTS/ASR model with re-ranking

| Method | CMOS |
|---|---|
| Speech-T | 0 |
| Speech-T (Re-ranking) | +0.11 |

$$\mu = \frac{1}{N} \sum_{k=1}^{N} m_k$$

$$CIs = [\mu - 1.96\frac{\sigma}{N}, \mu + 1.96\frac{\sigma}{N}]$$

where $\sigma$ is the standard deviation of the scores collected.

Table 9: MOS criteria.

| Voice Quality | Excellent | Good | Fair | Poor | Bad |
|---|---|---|---|---|---|
| Rating | 5 | 4 | 3 | 2 | 1 |

## A.4 50 Particularly Hard Sentences

The 50 particularly hard sentences mentioned in Section 4.2 are listed below:

01. a

02. b

03. c

04. H

05. I

06. J

07. K

08. L

09. 22222222 hello 22222222

10. S D S D Pass zero - zero Fail - zero to zero - zero - zero Cancelled - fifty nine to three - two - sixty four Total - fifty nine to three - two -

11. S D S D Pass - zero - zero - zero - zero Fail - zero - zero - zero - zero Cancelled - four hundred and sixteen - seventy six -

12. zero - one - one - two Cancelled - zero - zero - zero - zero Total - two hundred and eighty six - nineteen - seven -

13. forty one to five three hundred and eleven Fail - one - one to zero two Cancelled - zero - zero to zero zero Total -

14. zero zero one , MS03 - zero twenty five , MS03 - zero thirty two , MS03 - zero thirty nine ,

15. 1b204928 zero zero zero zero zero zero zero zero zero zero zero zero zero one seven ole32

16. zero zero zero zero zero zero zero zero two seven nine eight F three forty zero zero zero zero zero six four two eight zero one eight

17. c five eight zero three three nine a zero bf eight FALSE zero zero zero bba3add2 - c229 - 4cdb -

18. Calendaring agent failed with error code 0x80070005 while saving appointment .

19. Exit process - break ld - Load module - output ud - Unload module - ignore ser - System error - ignore ibp - Initial breakpoint -

20. Common DB connectors include the DB - nine , DB - fifteen , DB - nineteen , DB - twenty five , DB - thirty seven , and DB - fifty connectors .

21. To deliver interfaces that are significantly better suited to create and process RFC eight twenty one , RFC eight twenty two , RFC nine seventy seven , and MIME content .

22. int1 , int2 , int3 , int4 , int5 , int6 , int7 , int8 , int9 ,

23. seven _ ctl00 ctl04 ctl01 ctl00 ctl00

24. Http0XX , Http1XX , Http2XX , Http3XX ,

25. config file must contain A , B , C , D , E , F , and G .

26. mondo - debug mondo - ship motif - debug motif - ship sts - debug sts - ship Comparing local files to checkpoint files ...

27. Rusbvts . dll Dsaccessbvts . dll Exchmembvt . dll Draino . dll Im trying to deploy a new topology , and I keep getting this error .

28. You can call me directly at four two five seven zero three seven three four four or my cell four two five four four four seven four seven four or send me a meeting request with all the appropriate information .

29. Failed zero point zero zero percent < one zero zero one zero zero zero zero Internal . Exchange . ContentFilter . BVT ContentFilter . BVT_log . xml Error ! Filename not specified .

30. C colon backslash o one two f c p a r t y backslash d e v one two backslash oasys backslash legacy backslash web backslash HELP

31. src backslash mapi backslash t n e f d e c dot c dot o l d backslash backslash m o z a r t f one backslash e x five

32. copy backslash backslash j o h n f a n four backslash scratch backslash M i c r o s o f t dot S h a r e P o i n t dot

33. Take a look at h t t p colon slash slash w w w dot granite dot a b dot c a slash access slash email dot

34. backslash bin backslash premium backslash forms backslash r e g i o n a l o p t i o n s dot a s p x dot c s Raj , DJ ,

35. Anuraag backslash backslash r a d u r five backslash d e b u g dot one eight zero nine underscore P R two h dot s t s contains

36. p l a t f o r m right bracket backslash left bracket f l a v o r right bracket backslash s e t u p dot e x e

37. backslash x eight six backslash Ship backslash zero backslash A d d r e s s B o o k dot C o n t a c t s A d d r e s

38. Mine is here backslash backslash g a b e h a l l hyphen m o t h r a backslash S v r underscore O f f i c e s v r

39. h t t p colon slash slash teams slash sites slash T A G slash default dot aspx As always , any feedback , comments ,

40. two thousand and five h t t p colon slash slash news dot com dot com slash i slash n e slash f d slash two zero zero three slash f d

41. backslash i n t e r n a l dot e x c h a n g e dot m a n a g e m e n t dot s y s t e m m a n a g e

42. I think Rich's post highlights that we could have been more strategic about how the sum total of XBOX three hundred and sixtys were distributed .

43. 64X64 , 8K , one hundred and eighty four ASSEMBLY , DIGITAL VIDEO DISK DRIVE , INTERNAL , 8X ,

44. So we are back to Extended MAPI and C++ because . Extended MAPI does not have a dual interface VB or VB .Net can read .

45. Thanks , Borge Trongmo Hi gurus , Could you help us E2K ASP guys with the following issue ?

46. Thanks J RGR Are you using the LDDM driver for this system or the in the build XDDM driver ?

47. Btw , you might remember me from our discussion about OWA automation and OWA readiness day a year ago .

48. empidtool . exe creates HKEY_CURRENT_USER Software Microsoft Office Common QMPersNum in the registry , queries AD , and the populate the registry with MS employment ID if available else an error code is logged .

49. Thursday, via a joint press release and Microsoft AI Blog, we will announce Microsoft's continued partnership with Shell leveraging cloud, AI, and collaboration technology to drive industry innovation and transformation.

50. Actress Fan Bingbing attends the screening of 'Ash Is Purest White (Jiang Hu Er Nv)' during the 71st annual Cannes Film Festival