# OpenReview forum: "Speech-T: Transducer for Text to Speech and Beyond"
_NeurIPS.cc/2021/Conference — NeurIPS 2021 Poster_

### Official Review · Reviewer_kkqQ · 2021-07-14

**Rating:** 6
**Confidence:** 5

**Summary:**

Motivation: Transducer for TTS, streaming

Model:
Uses convolution + self-attention on text input (either causal/auto-regressive for streaming, or full context).
Convolution + auto-regressive self-attention for mel-spectogram (audio features).

Very nice idea to use transducer for TTS, to separate blank from the emissions (but not new).

Also, as one further extension: Single combined model for both ASR + TTS.


**Limitations And Societal Impact:**

Does not apply.

**Main Review:**

The application of the transducer to TTS is novel (as far as I know).

> For the first challenge, we design a lazy forward algorithm that separates transition prediction from mel-spectrogram prediction ...

This separation of blank (separation of transition + emission) is also done in ["Hybrid Autoregressive Transducer (HAT)"](https://arxiv.org/abs/2003.07705) and ["A New Training Pipeline for an Improved Neural Transducer"](https://arxiv.org/abs/2005.09319).
So this separation is not really new (as it sounds from the text). But the application of this idea to TTS is new. And using it as an approximation for the dynamic programming (ignoring the emission prob) is also new (as far as I know). However, I think for the idea of the separation itself (transition + emission), the corresponding earlier work should be cited, which did the same kind of separation.

There are some problems in this work. Esp there are potential flaws with the loss function. The dependence on an external alignment is another problem. See below for more details.

The code should be published as well.

This work is unfortunately not quite complete, and seems more work-in-progress (see below for further comments on this).

There are some open questions, so the clarity could be improved. (See below.)

The overall organization could be better. It seems as if the work first started with the non-streaming model. And then in a later section, the streaming model is introduced. And later, the combined ASR+TTS model. The earlier sections are written in a way that completely ignores the later sections.

This work can be very significant. However, it should publish the code to better allow others to build on this. And the current work should be completed further.

---

Some random further comments:

> ... existing solutions for streaming TTS [14] mostly rely on the argmax operation of the encoder-decoder attention values to decide the end of speech frame corresponding to an input word, which is not accurate and robust. Transducer can naturally support streaming inputs ...

But the Transducer would also use argmax during inference time for the transition? I don't exactly understand the argument about the argmax.


> In both TTS and ASR, we use T and U to represent the length of text and speech sequence respectively, and use X and Y to represent the input and output sequence respectively.

T and U is clear then. But X and Y is unclear. Input is speech in ASR, and input is text in TTS. Or does X always mean speech?


> As shwon in Figure ...

Typo.


> A difference between the two encoders is that the speech encoder adopts causal self-attention and convolution to only see the previous context to support the autoregressive prediction in joint network.

I thought the text encoder also needs to be auto-regressive to support streaming? Otherwise you must know the whole text in advance, and this is not streaming.Or I assume this streaming aspect comes later? It is a bit confusing to read this here, as this sounds as if it would apply in general, not only for the initial non-streaming experiments.


> Loss function of our proposed Speech-TTS

The probability is not in log space? Why? This is inconsistent to the original transducer loss.

The formula is also wrong in that the loss should only use alpha(T,U) (because that recursively includes the whole probability). This is strange, and wrong, and again not the same as in the original transducer. Was it used in this incorrect way?

The approach of the "lazy" calculation using the emission binary prob only is fine, and actually very nice. But maybe it is applied incorrectly (as argued above).

For the comparison to the original loss (Section 4.3), I'm not sure if the emission probability is used correctly. The mel-spectogram prediction can be formulated in a probabilistic way as in the mixture density network approach. This would use a simple Gaussian, not even a mixture, not even prediction the variance, just the mean, and then you actually almost have that, except that the loss would be different (not L1 but similar as scaled L2).

> Figure 3

This figure is strange in that $T > U$ (i.e. num text labels is longer than num speech frames), which is usually not the case, but you have $U >> T$.

> Section 3.3, Alignment Constraint

So it turns out this needs good external alignments, from other external models, and otherwise does not work so well (Table 5). This is a big shortcoming. The model should be able to produce good alignments by itself.

The question is why. Maybe because the loss is actually flawed (as argued before)?

If it needs existing good alignments, a simpler framewise loss along the path could be used, to not calculate the joint network for the whole lattice. Similar as in ["A New Training Pipeline for an Improved Neural Transducer"](https://arxiv.org/abs/2005.09319).

---

> Did you report error bars (e.g., with respect to the random seed after running experiments multiple times)? [N/A] The quality of a speech synthesis model is relatively stable. Besides, MOS test by multiple judgers on multiple sentences can provide mean and variance of voice quality.

In my experience, it is not stable at all (at least attention-based models). The MOS test for a single model but multiple sentences does not tell anything about the variance due to different random seeds. I think error bars via different random seeds would be an interesting addition here.

> Did you include the code, data, and instructions needed to reproduce the main experimental results (either in the supplemental material or as a URL)? [Yes] See Section 4.1.

This seems to be incorrect. Section 4.1 just explains what the code was based on, and the modifications made. It seems that the new code will not be included. This is a serious problem. This code must be published alongside this work.

---

There is a lot of potential in this work. The approach and idea of using the transducer for TTS, or even ASR and TTS combined, is very nice.

Unfortunately, it is still somewhat incomplete. There are many open questions. There seem to be some errors (e.g. flaws in the loss function). There is the shortcoming of needing an external alignment, which is unexpected, and should be studied and understood better, why this is needed (maybe because of the flaws in the loss function). The new proposed loss also should be compared to the standard transducer loss, which is missing, but important for analysis and better understanding. Also, the new code must be published.

If you want to make this idea / proposal of your model public right away, I would suggest to publish it on Arxiv now anyway, and maybe submit it to some workshop, where such work-in-progress papers are more appropriate.


**Time Spent Reviewing:**

2

---

> ### Author Response · Authors · 2021-08-10
> **Response to Reviewer kkqQ (Part 1)**
>
> Thank you for your comments and suggestions!
>
> **[Q1: For the idea of the separation itself (transition + emission), the corresponding earlier work should be cited.]**
>
> A1: Thanks for your suggestion. We will cite these two related work in the new version of our paper. While they also separate the blank (separation of transition + emission), there are fundamental difference between our proposed method and their work:
>
> (1) Their transition is modeled by a Bernoulli distribution φ(t, u), and their emission is modeled by (1-φ(t, u))p(t, u), where p(t, u) represents the probability of target token prediction. They still incorporate the transition probability φ(t, u) and token prediction probability p(t, u) together to calculate the probability lattice, suffering from the trade-off problem between the two probabilities if applying to TTS.
>
> (2) In our lazy forward algorithm, we only use transition and non-transition probability to calculate the lattice probability, which can naturally avoid the trade-off problem. The lattice probability is further used to weight the L1 mel-spectrogram loss as the final loss.
>
>
>
>
>
> **[Q2: Regarding to the loss function of our proposed Speech-T.]**
>
> A2: We reply to your comments one by one as follows.
>
> (1) “The probability is not in log space? Why? This is inconsistent to the original transducer loss.”
>
> - The loss function we proposed is not the same as the original transducer loss. The original transducer loss is for ASR, which predicts text tokens by classification and usually uses log space to convert the probability multiplication into summation. However, our transducer loss is for TTS, which predicts mel-spectrogram by regression (e.g., L1 loss), which does not need to conduct in log space.
>
>
> (2) “The formula is also wrong in that the loss should only use alpha(T,U) (because that recursively includes the whole probability). This is strange, and wrong, and again not the same as in the original transducer. Was it used in this incorrect way?’.
>
> - We are sorry but we disagree with you about this. The original transducer loss for ASR uses alpha (T, U) recursively to include all probabilities. However, in order to avoid the trade-off between the emission and transition probability when calculating the output probability lattice in Transducer for TTS, we propose a lazy forward algorithm in our Speech-T. Specifically, the output probability lattice is lazy, which only calculates the probability of transition/non-transition to get the forward variable alpha(t, u), without taking the probability of mel-spectrogram prediction into account. In this way, the forward variable only depicts the transition dynamics of the lattice, but not the emission probability. Therefore, we cannot only use alpha(T, U) to derive the loss function. Instead, as shown in Equation 4, we weight the mel-spectrogram prediction with the probability of emission at each node by taking the forward variable into account.
>
> (3) “For the comparison to the original loss (Section 4.3), I'm not sure if the emission probability is used correctly. The mel-spectogram prediction can be formulated in a probabilistic way as in the mixture density network approach. This would use a simple Gaussian, not even a mixture, not even prediction the variance, just the mean, and then you actually almost have that, except that the loss would be different (not L1 but similar as scaled L2).”
>
> - The emission probability is used correctly. We explain as follows.
>   - A key challenge when designing Transducer for TTS is that it is difficult to trade off the emission and transition probability when calculating the output probability lattice in Transducer. This tradeoff problem does not exist in ASR Transducer, since the blank token is added into the token vocabulary in ASR, which means that target token prediction (emission) and blank token prediction (transition) are modeled in a unified probability distribution through the softmax on all the vocabulary tokens. However, the mel-spectrogram in TTS is continuous and mel-spectrogram prediction (emission) cannot be modeled by a joint probability distribution with blank token prediction (transition).
>   - Therefore, the key is not to model the mel-spectrogram prediction in a probabilistic way or a non-probabilistic way like L1 loss, but how to balance the transition probability of a discrete random variable and mel-spectrogram prediction of a continuous vector when calculating the output probability lattice. Table 4 shows the ablation study where the GMM-based method uses mixed Gaussian probability to model the mel-spectrogram. However, since it cannot well trade off the transition probability and emission probability, it results in much worse quality than our proposed lazy forward algorithm.
>   - The key of our lazy forward algorithm is that we only use transition/non-transition to derive the probability lattice. In this way, we avoid the trade-off problem between the transition and emission probabilities, since we do not use emission probability (mel-spectrogram prediction) in calculating the probability lattice. Then, we use the probability in the lattice to weight the mel-spectrogram prediction loss, as shown in Equation 4.
>   - You may argue that by modeling mel-spectrogram as Gaussian probability model instead of L1 loss based on our proposed lazy forward algorithm, the model output is at least a true probability instead of a pseudo probability (negative L1 loss), which may be helpful. However, we also tried to use Gaussian probability to model the mel-spectrogram instead of L1 loss in our proposed lazy forward algorithm (change the L1 loss in Equation 4 with negative Gaussian probability by predicting mean and variance in the joint network) before paper submission. We found there is no difference in voice quality between Gaussian probability and L1 loss. To keep simplicity, we use L1 loss in our paper, as it is commonly used in mel-spectrogram prediction. We will add the above discussion in our new version of paper.
>
>
>
>
>
> **[Q3: “The code should be published as well.”]**
>
> A3: Thanks for the suggestion! We attach the code in this anonymous link (https://anonymous.4open.science/r/speech_t-CA83/). We will release it once the paper is open to public.
>
>
>
> **[Q4: About the paper organization.]**
>
> A4: We describe the paper organization as follows:
>
> (1) We first describe the basic formulation of Speech-T which works in non-streaming mode, and then introduce its streaming variation with k-lookahead mechanism. Streaming and non-streaming models have no big difference in the key design of Transducer (i.e., lazy forward algorithm and alignment constraint). Besides, k=infinity equals to non-streaming mode.
>
> (2) Combining TTS and ASR is an extension of Speech-T, so we formulate it in a separate section.
>
> (3) We have introduced the details of these organizations in the end of Section 1, and the beginning of Section 3 and Section 4. Anyway, we will improve the organization of the paper and make it easier to follow.
>
>
>
> **[Q5: About the difference between argmax operation in neural transducer and encoder-decoder attention framework.]**
>
> A5: Although both frameworks use argmax operation for transition, they are different:
>
> (1) In Transducer for TTS, the blank prediction is modeled as a binary classification, where the argmax operation is used to decide transit or not. No matter transition or non-transition, the alignment is still monotonic.
>
> (2) In encoder-decoder attention for TTS, to decide a transition, the argmax position should be one position ahead of the previous attention position (argmax position) due to the monotonicity in text-speech alignment. However, since the argmax operation is conducted over all possible phoneme positions, the argmax position may not be exact one position ahead, which causes word skipping (more than one position ahead) or repeating (current position or position behind).
>
>
>
> **[Q6: Regarding to the meaning of X and Y in Footnote 1 in Section 3.1.]**
>
> A6: For the consistency of the notations in both TTS and ASR, we always use T and U to represent the length of text and speech sequence, respectively, and use X and Y to represent the input and output sequence respectively. In ASR, X is the input speech sequence, and Y is the output text sequence; In TTS, X is the input text sequence, and Y is the output speech sequence Y.
>
>
>
> **[Q7: Does text encoder also need autoregressive to support streaming?]**
>
> A7: From Section 3.1 to Section 3.3, we still formulate the basic setting with non-streaming text input, and then describe the variant with streaming input in Section 3.4. Therefore, the text encoder can see the whole sentence in the basic setting, but can only see previous tokens and k future tokens in the streaming setting with k-lookahead mechanism as described in Section 3.4. We will make it clear in the paper to avoid confusion.
>
>
>
>
>
> **[Q8: About Figure 3. ]**
>
> A8: Thanks for pointing out the issue. We will re-draw the figure to make U larger than T in our new version of paper.

---

> > ### Author Response · Authors · 2021-08-10
> > **Response to Reviewer kkqQ (Part 2)**
> >
> > **[Q9: About alignment constraints.]**
> >
> > A9: We answer this question from the following aspects:
> >
> > (1)  We need to clarify that our loss has no problem as we explain in (Q2 and A2).
> >
> > (2)  The alignments between text and speech are more difficult in TTS than that in ASR:
> >
> > - First, let’s consider the alignments from the perspective of target sequence. Each target token can be aligned to multiple source speech frames in ASR, while each target frame can be aligned to only a single token in TTS. When a few frames of a token are wrongly aligned, it is likely the predicted target token in ASR is still correct, since there are more correctly aligned frames to this target token. However, in TTS, if a few predicted target speech frames are misaligned with the source token, it will lead to skipping or repeating errors [10,11] and seriously hurt the quality of synthesized speech.
> >
> > - Second, when predicting current mel-spectrogram in TTS, previous mel-spectrograms can provide enough information as they are very similar due to the continuity between consecutive mel-spectrograms, which may lead to copying the previous frame instead of learning information from the text encoder and thus harms alignment learning [7,8,11]. However, the adjacent target tokens in ASR are usually not similar and thus this problem is not serious in ASR.
> >
> >
> >
> > (3)  Although we leverage external alignment module, it is a well-developed and open-source tool (https://github.com/MontrealCorpusTools/Montreal-Forced-Aligner) that is easy to use. Relying this kind of alignment tool for TTS modeling is a common practice [1,2,3,4,5].
> >
> > (4)  In order to avoid the trade-off between the emission and transition probability when calculating the output probability lattice in Transducer, we only use transition/non-transition probability to calculate the probability lattice, without eagerly taking the mel-spectrogram prediction into account. To the best of our knowledge, Speech-T is the first work to successfully design a neural Transducer model for TTS with high-voice quality, which is non-trivial. In order to ease the alignment learning, we leverage an external alignment tool. However, reducing the dependency of Speech-T on external alignment tool is definitely a promising research direction. We leave it to future work.
> >
> > (5) About “a simpler framewise loss along the path could be used”
> >
> > - In the alignment constraint in Section 3.3, to ensure the flexibility in the constraint, we extend the alignment path by τ frames in both the left and right along the horizonal axis to form a diagonal and banded region, as shown in the green area in Figure 3. While τ is set to 1, it is equivalent to the framewise mel-spectrogram loss. However, we also tried this setting before paper submission, and found it results in much worse quality than our setup in the experiment (τ=10). Specifically, we found that setting τ=1 can easily encounter issues such as getting stuck for many steps in some phonemes without jumping to the next phoneme. Although this framewise loss has been proved to be effective in ASR, it cannot work in TTS due to the distinctive characteristics of alignments in TTS (as we discuss in point (2) in this Q9 and A9). In this way, this framewise loss for TTS requires highly accurate alignment than that for ASR, which cannot be satisfied by the external alignment tool we used (note that the alignment tool we used is accurate enough for other purposes but not for this framewise loss).
> >
> >
> >
> >
> >
> > **[Q10: About error bars.]**
> >
> > A10: Thanks for your suggestion! To the best of our knowledge, most mainstream TTS works [1-11] usually choose the best checkpoint according to the validation performance and then use it for MOS test. In this way, the chosen checkpoint is used to represent the performance of this model. However, your suggestion is very interesting. We will conduct dedicated research to study the stability of TTS models.
> >
> >
> >
> > **[Q11: “The new proposed loss also should be compared to the standard transducer loss, which is missing, but important for analysis and better understanding.”]**
> >
> > A11: Actually, we have done this in Table 4. The default algorithm is the standard transducer loss. As can be seen, it cannot generate any reasonable speech, which demonstrates the effectiveness of our proposed Speech-T. If you mean the standard transducer loss with mel-specotrogram prediction based on Gaussian probability, please refer to point (3) in Q2 and A2.
> >
> >
> >
> > **[Q12: “This work is unfortunately not quite complete, and seems more work-in-progress.” ]**
> >
> > A12: The main goal of our paper is to design a novel method (Speech-T) that leverages Transducer for TTS. To this end, we propose a lazy forward algorithm to avoid the trade-off problem between emission and transition when calculating the probability lattice, and an attention constraint to ensure the alignment learning. Besides, we also investigate the streaming setting of Speech-T. We have detailed method formulation. Experiment results and ablation studies demonstrate the advantages of our method and the effectiveness of each key design. As a further extension, we investigate the joint TTS/ASR modeling with Speech-T. We think our work is complete, from motivation, to method formulation, to experiment studies, to further extension.
> >
> > Based on all the responses above, we hope you can have a better understanding on our paper, and sincerely wish you can reconsider your judgement and rating. If any more information and clarification are needed, feel free to ask us. Thanks very much for your efforts in reviewing our paper!
> >
> >
> >
> > > *References:*
> > >
> > > *[1] Yu C, Lu H, Hu N, et al. Durian: Duration informed attention network for multimodal synthesis[J]. arXiv preprint arXiv:1909.01700, 2019.*
> > >
> > > *[2] Li N, Liu Y, Wu Y, et al. Robutrans: A robust transformer-based text-to-speech model[C]//Proceedings of the AAAI Conference on Artificial Intelligence. 2020, 34(05): 8228-8235.*
> > >
> > > *[3] Ren Y, Hu C, Tan X, et al. FastSpeech 2: Fast and High-Quality End-to-End Text to Speech[C]//International Conference on Learning Representations. 2020.*
> > >
> > > *[4] Elias I, Zen H, Shen J, et al. Parallel Tacotron: Non-autoregressive and controllable TTS[C]//ICASSP 2021-2021 IEEE International Conference on Acoustics, Speech and Signal Processing (ICASSP). IEEE, 2021: 5709-5713.*
> > >
> > > *[5] Shen J, Jia Y, Chrzanowski M, et al. Non-attentive tacotron: Robust and controllable neural TTS synthesis including unsupervised duration modeling[J]. arXiv preprint arXiv:2010.04301, 2020.*
> > >
> > > *[6] Naihan Li, Shujie Liu, Yanqing Liu, Sheng Zhao, and Ming Liu. Neural speech synthesis with transformer network. In Proceedings of the AAAI Conference on Artificial Intelligence, volume 33, pages 6706–6713, 2019.*
> > >
> > > *[7] Y. Wang, R. Skerry-Ryan, D. Stanton, Y. Wu, R. J. Weiss, N. Jaitly, Z. Yang, Y. Xiao, Z. Chen, S. Bengio, Q. Le, Y. Agiomyrgiannakis, R. Clark, and R. A. Saurous, “Tacotron: Towards end-to-end speech synthesis,” in Proc. Interspeech, Aug. 2017, pp. 4006–4010.*
> > >
> > > *[8] Shen J, Pang R, Weiss R J, et al. Natural tts synthesis by conditioning wavenet on mel spectrogram predictions[C]//2018 IEEE International Conference on Acoustics, Speech and Signal Processing (ICASSP). IEEE, 2018: 4779-4783.*
> > >
> > > *[9] Ren Y, Ruan Y, Tan X, et al. FastSpeech: fast, robust and controllable text to speech[C]//Proceedings of the 33rd International Conference on Neural Information Processing Systems. 2019: 3171-3180.*
> > >
> > > *[10] W. Ping, K. Peng, A. Gibiansky, S. O. Arik, A. Kannan, S. Narang, and J. Miller, “Deep Voice 3: Scaling Text-to-Speech with Convolutional Sequence Learning,” in International Conference on Learning Representations - ICLR, vol. 79, 2018, pp. 1094–1099.*
> > >
> > > *[11] Chen M, Tan X, Ren Y, et al. MultiSpeech: Multi-Speaker Text to Speech with Transformer[J]. Proc. Interspeech 2020, 2020: 4024-4028.*

---

> > > ### Comment · Reviewer_kkqQ · 2021-08-29
> > > **Response**
> > >
> > > Thank you for the detailed response. And also for publishing the code, which I really appreciate.
> > >
> > > I increased my score. I think this work is fine for acceptance, when my concerns are reasonably addressed.

---

### Official Review · Reviewer_GhAx · 2021-07-15

**Rating:** 6
**Confidence:** 4

**Summary:**

This paper proposed the use of transformer based transducers for the text-to-speech (TTS) task. Different from transducer based automatic speech recognition (ASR) systems which utilize a blank symbol at the output, MFCC prediction in TTS cannot accommodate for that. To solve this problem, the authors propose to use a lazy forward algorithm to compute the likelihood of the output. In addition, they add a diagonal constraint to achieve text-to-speech alignments that lie close to the diagonal of the alignment lattice. Even though the MOS score is not better than the baseline transformerTTS model, the results show reduction in the sentence errors. The paper also shows preliminary results on combining ASR and TTS in the same Speech-T model and shows slight reduction in the phoneme error rate with the joint model as compared to ASR-only model.

**Ethical Concerns:**

From the paper, it was not clear how the speaker variability will be handled in the model. Is there an analysis of how male vs female speech are generated and which ones have better quality?

**Limitations And Societal Impact:**

MOS scores in Table 1 make the quality of the output speech slightly questionable. However, quantitative analysis show that it is better than the baseline. Some questions to further explore might be if we can by-pass external modules such as grapheme-to-phoneme conversion and vocoder and achieve grapheme to time-domain speech signal synthesis.


**Main Review:**

Originality: This paper introduces tricks such as the lazy forward algorithm and the diagonal constraints so that transducers can be used for the TTS task successfully. The paper refers to a previous study on the use of transducers for TTS and tries to address the weaknesses of that model in this paper.

Quality: The method seems technically sound. Although qualitative results not significantly better than the baseline method, quantitative performance of Speech-T is better than the baseline model. Section 4.3 provides a limited but sufficient analysis of the effect of individual components.
One comment regarding the joint ASR and TTS experiment is that the ASR output is not performing ASR in the conventional sense, it rather performs a next phoneme prediction task. It is true that it takes speech and outputs a phoneme label but is slightly different than the sequence to sequence speech-to-text task.

Clarity: The paper is clearly written, except a few typos.
Line 120: nagative, Line 328: accruate, Line 338: strcutrue, Line 339: reusults


Significance: The paper provides practical methods for the use of transducers for the TTS task which would provide a tool for the TTS community. The joint ASR+TTS training could be interesting to the ASR community as well.

+++ Updates after author-rebuttal:
I have read the reviews by other reviewers and the author-rebuttal. A few comments on the novelty regarding the use of by Reviewer kkqQ make me re-evaluate the novelty score of the paper. This slightly lowers my original score. Also, the background information needs an update as mentioned by Reviewer fXgK.

**Time Spent Reviewing:**

1.5

---

> ### Author Response · Authors · 2021-08-10
> **Response to Reviewer GhAx**
>
> Thank you for your feedback and comments!
>
> **[Q1: Regarding to phoneme prediction task.]**
>
> A1: Thanks for your suggestion. Our main focus in this paper is to propose a neural Transducer model for TTS. To achieve this, we propose a lazy forward algorithm and an alignment constraint to ensure the quality. As an extension, we apply our proposed method to modeling both TTS and ASR in a joint model. We choose phoneme sequence as the target to verify the performance, which is mainly used to demonstrate the great potential of our proposed method. We will use text sequence as the target in future work.
>
>
>
> **[Q2: Is it possible to bypass grapheme-to-phoneme conversion and vocoder, and achieve grapheme to time-domain speech signal synthesis?]**
>
> A2: Thanks for your suggestion. Yes, bypassing grapheme-to-phoneme conversion and vocoder is the ultimate goal of TTS. While there are some fully end-to-end models that directly convert grapheme sequence into waveform, the mainstream TTS models still use a three-stage cascaded pipeline: 1) text analysis, e.g., grapheme-to-phoneme; 2) acoustic model, converting phoneme sequence into mel-spectrogram sequence; 3) vocoder, converting mel-spectrogram sequence into waveform. How to design better acoustic model for phoneme to mel-spectrogram conversion is still the most challenging problem in TTS. Nevertheless, designing Transducer model for fully end-to-end TTS is attractive. We will definitely consider it in our future work.
>
>
>
> **[Q3: How the speaker variability will be handled in the model?]**
>
> A3: Our Speech-T is verified in a single-speaker TTS dataset, similar to most research work on TTS model design [1,2,3,4]. However, we can easily extend it to multiple speakers, by using speaker look-up table or speaker encoder [5,6,7] to model the characteristics of speakers. We can add the speaker embedding to the input of the speech encoder to ensure Speech-T can model and generate speech in multiple speakers.
>
>
> > *References:*
> >
> > *[1] Wang Y, Skerry-Ryan R J, Stanton D, et al. Tacotron: Towards End-to-End Speech Synthesis[J]. Proc. Interspeech 2017, 2017: 4006-4010.*
> >
> > *[2] Shen J, Pang R, Weiss R J, et al. Natural tts synthesis by conditioning wavenet on mel spectrogram predictions[C]//2018 IEEE International Conference on Acoustics, Speech and Signal Processing (ICASSP). IEEE, 2018: 4779-4783.*
> >
> > *[3] Li N, Liu S, Liu Y, et al. Neural speech synthesis with transformer network[C]//Proceedings of the AAAI Conference on Artificial Intelligence. 2019, 33(01): 6706-6713.*
> >
> > *[4] Ren Y, Ruan Y, Tan X, et al. FastSpeech: fast, robust and controllable text to speech[C]//Proceedings of the 33rd International Conference on Neural Information Processing Systems. 2019: 3171-3180.*
> >
> > *[5] A. Gibiansky, S. Arik, G. Diamos, J. Miller, K. Peng, W. Ping, J. Raiman, and Y. Zhou, “Deep voice 2: Multi-speaker neural text- to-speech,” in Advances in neural information processing systems, 2017, pp. 2962–2970.*
> >
> > *[6] W. Ping, K. Peng, A. Gibiansky, S. O. Arik, A. Kannan, S. Narang, and J. Miller, “Deep Voice 3: Scaling Text-to-Speech with Convolutional Sequence Learning,” in International Conference on Learning Representations - ICLR, vol. 79, 2018, pp. 1094–1099.*
> >
> > *[7] Chen M, Tan X, Ren Y, et al. MultiSpeech: Multi-Speaker Text to Speech with Transformer[J]. Proc. Interspeech 2020, 2020: 4024-4028.*

---

### Official Review · Reviewer_fXgK · 2021-07-20

**Rating:** 5
**Confidence:** 5

**Summary:**

This is a very interesting application of the Neural Transducer framework that is quite popular for end-to-end ASR to the problem of TTS. The authors adopt the dynamic programming sum over many possible alignments of text and audio to minimize a TTS-oriented L1 frame prediction loss. The details they propose are reasonable; though not the only way forward, the proposal is to decouple the alignment path probability from frame prediction probabilities, while nonetheless multiplying alignment path probability by the L1 prediction loss. Good results are presented compared to a strong Transformer TTS baseline, though I am not sure whether that baseline is itself as strong as the best WaveNet or Tacotron results. Furthermore, a unified TTS/ASR system is described sharing parameters of the acoustic/text encoders, predicting both feature frames for TTS and next symbols for ASR.

**Limitations And Societal Impact:**

I don't see any issues here that stand out in this regard.

**Main Review:**

This is a reasonable proposal whose time has come; it is the natural TTS counterpart of the plethora of ASR work done with Neural Transducers.

Overall, though the presentation is on the whole clear, the work suffers from some poor writing and grammatical errors, and from a slight lack of historical context.

The authors might want to frame the work in a more standard framework for auto-regressive generative modeling of speech. The proposal is very much the successor of HMM-based speech synthesis, given the independence assumptions of the Neural Transducer that allow the use of the forward summation of path scores. In particular, they seem to struggle with the concept of the path alignment probability given their use of an L1 loss, but that can be seen as a special case of a single gaussian probability model with shared radial variance. Eq (4) uses path probabilities to weight the L1 prediction loss, but the path probabilities themselves are only dependent on a binary transition probability model. So Eq. (4) is a hybrid loss with no clear probabilistic interpretation other than "weighted L1 loss". Instead, the loss could be formulated along the lines of HMM-based TTS as a generative model of the audio, and indeed as a deep generative model along the lines proposed e.g. by

M. Schuster , "On supervised learning from sequential data with applications for speech recognition", PhD Thesis, 1999.

H. Zen and A. Senior, "Deep mixture density networks for acoustic modeling in statistical parametric speech synthesis", 2014.

van den Oord et al. "Wavenet", 2016

H. Zen, "Generative Model-Based Text-to-Speech Synthesis", 2018

Given the Neural Transducer model's simplifying independence assumptions, a generative model along those lines can then employ the highly convenient/efficient forward path summation (= dynamic programming).

Specific comments:

Re: Section 2, background: it is confusing how they have copied the Alex Graves RNN-T framework, but inverted some aspects. They use T to denote the length of the text sequence and U the length of the audio utterance, while it is the other way around in the original Graves work. Thus Eq. (2) is confusing in its non-standard sense of what is a transition to a new symbol and what is remaining in the current symbol.

Re: "Computing Equation (1) by summing all possible alignments naively is computationally intractable. Therefore, Graves [7] introduced an efficient forward algorithm to handle this problem": the forward algorithm was proposed decades before Graves' work and used in ASR for 20-25 years before Graves, see e.g.

Kai Fu Lee, Automatic Speech Recognition -- The Development of the SPHINX System, 1989.

L. R. Rabiner and B.-H. Juang, Fundamentals of Speech Recognition, PTR Prentice-Hall, 1993.

Re: phrasing of "Application of Transducer", in many places. "Transducer" is a general term that is particular well known in the speech and language community from the Weighted Finite State Transducer work by Riley & Mohri, and others. I think it should be "Neural Transducer" -- as an abbreviation for the specific model proposed by Alex Graves.

Re: "1) It is hard to trade off the probability of transition and emission (mel-spectrogram prediction) when calculating the output probability lattice, since mel-spectrogram and transition predictions are modeled separately (mel-spectrogram prediction using regression, while transition prediction using binary classification)" : see my previous comments

Re: "2) The alignments between speech and text in TTS are more difficult to learn than those in ASR" : how is the alignment problem fundamentally different between TTS and ASR?

Re: constrained alignments and monotonic RNN-T, see the specifically monotonic constraints (no purely vertical transitions) proposed in https://ieeexplore.ieee.org/document/9003822, "Monotonic Recurrent Neural Network Transducer and Decoding Strategies" by  Anshuman Tripathi; Han Lu; Hasim Sak; Hagen Soltau, 2019.



**Time Spent Reviewing:**

2 hours

---

> ### Author Response · Authors · 2021-08-10
> **Response to Reviewer fXgK**
>
> Thank you for your feedback and comments!
>
> **[Q1: About struggling with the concept of the path alignment probability given their use of an L1 loss.]**
>
> A1: We address your concerns one by one as follows.
>
> (1)	“The proposal is very much the successor of HMM-based speech synthesis, given the independence assumptions of the Neural Transducer that allow the use of the forward summation of path scores.”
>
> - Although HMM, CTC, and Transducer rely on some common methods such as forward algorithm, they are totally different models. Introducing the analogies between HMM and Transducer is very interesting, but may not help to understand exactly the motivation and logic of our proposed method.
>
> (2)	“In particular, they seem to struggle with the concept of the path alignment probability given their use of an L1 loss” and “Given the Neural Transducer model's simplifying independence assumptions, a generative model along those lines can then employ the highly convenient/efficient forward path summation (= dynamic programming).”
>
> - The key problem when designing Transducer for TTS is that it is difficult to trade off the emission and transition probability when calculating the output probability lattice in Transducer. This tradeoff problem does not exist in ASR Transducer, since the blank token is added into the token vocabulary in ASR, which means that the emission (token prediction) and transition (blank prediction) are modeled in a unified probability distribution through a softmax on all the vocabulary tokens. However, in TTS, the transition (blank token prediction) is modeled as a binary classification, and the emission (mel-spectrogram prediction) is usually modeled as regression or gaussian models since mel-spectrogram is continuous. Unlike Transducer ASR, the transition and emission in TTS cannot be modeled by a joint probability distribution, which causes the trade-off problem.
> - Therefore, the key is not to model the mel-spectrogram prediction as a Gaussian probability or an L1 loss (use the negative L1 loss as the probability), but how to balance the transition probability and mel-spectrogram prediction term when calculating the output probability lattice. You can find the ablation study in Table 4 in our paper submission, where the GMM-based method uses mixed Gaussian probability to model the mel-spectrogram. However, because it cannot well trade off the transition and emission probabilities, it still results in much worse quality than our proposed lazy forward algorithm.
>
> (3) “ Eq (4) uses path probabilities to weight the L1 prediction loss, but the path probabilities themselves are only dependent on a binary transition probability model. So Eq. (4) is a hybrid loss with no clear probabilistic interpretation other than "weighted L1 loss” ”
>
> - The key of our lazy forward algorithm is that we only use transition/non-transition to derive the probability lattice. In this way, we avoid the trade-off problem between the transition and emission probabilities, since we do not use emission probability (mel-spectrogram prediction) in calculating the probability lattice. Then, we use the probability in the lattice to weight the mel-spectrogram prediction loss, as shown in Equation 4.
> - As long as we can solve the trade-off problem between the transition probability and mel-spectrogram prediction in calculating the probability lattice, our loss does not need to be fully probabilistic. Actually, we use the widely adopted L1 loss for mel-spectrogram prediction. Besides, our weighted L1 loss has clear interpretation: when making mel-spectrogram prediction at node (t, u), we need to consider how much probability can flow to node (t, u), i.e., alpha(t, u), and the probability of making a prediction instead of transition at node (t, u), i.e., 1-phi(t, u).
> - You may argue that by modeling mel-spectrogram with Gaussian probability instead of L1 loss based on our proposed lazy forward algorithm, the model output is at least a true probability instead of a pseudo probability (negative L1 loss), which may be helpful. We also tried to use Gaussian probability to model the mel-spectrogram instead of L1 loss in our proposed lazy forward algorithm (change the L1 loss in Equation 4 with negative Gaussian probability by predicting mean and variance in the joint network) before paper submission. We found there is no difference in voice quality between Gaussian probability and L1 loss. To keep simplicity, we use L1 loss in our paper. We will add the above discussions in updated version of the paper.
>
>
>
> **[Q2: About the notation in Equation 2.]**
>
> A2: There is no big difference from the original RNN-T work except for that we use T to represent the length of text sequence and U to represent that of speech sequence, i.e., we reverse the notation in original RNN-T since TTS is our focus in this paper. For the consistency of the notations in both TTS and ASR, as we said in Footnote 1 in Section 3.1, we always use T and U to represent the length of text and speech sequence, respectively, and use X and Y to represent the input and output sequence respectively. For example, in ASR, the input speech sequence X has length of U, and the output text sequence Y has length of T; In TTS, the input text sequence X has length of T, and output speech sequence Y has length of U.
>
>
>
> **[Q3: Regarding to ‘the forward algorithm was proposed decades before Graves work’.]**
>
> A3: Thanks for pointing out this issue. It is just a typo. By using ‘introduced’ in the sentence ‘Therefore, Graves [7] introduced an efficient forward algorithm’, we just mean ‘Graves [7] adopted a previous designed algorithm’, but not ‘designed a new algorithm’. We will make it clear in the new version of the paper.
>
>
>
> **[Q4: About the term ‘Neural Transducer’.]**
>
> A4: Thanks for your suggestion! We will use “Neural Transducer” in the new version of paper.
>
>
>
> **[Q5: How is the alignment problem fundamentally different between TTS and ASR?]**
>
> A5: There are several differences in the alignment problem between TTS and ASR:
>
> (1) First, considering the alignments from the perspective of target sequence, each target token can be aligned to multiple source speech frames in ASR, while each target frame can be only aligned to a single token in TTS. When there are some slight alignment errors, the predicted target token in ASR may be still correct, since there still exist some corrected aligned frames to this target token. However, in TTS, the predicted target speech frame may have skipping or repeating errors [3,4], since the source token is misaligned.
>
> (2) Second,  when predicting current mel-spectrogram in TTS, previous mel-spectrograms can provide enough information as they are very similar due to the continuity between consecutive mel-spectrograms, which may lead to simply copying the previous frame instead of learning information from the text encoder and thus harms alignment learning [1,2,3]. However, the adjacent target tokens in ASR are not similar and thus this problem is not serious in ASR.
>
>
>
> **[Q6: Constrained alignments and monotonic RNN-T.]**
>
> A6: We will add and discuss the reference in the new version of paper.
>
>
> > *References:*
> >
> > *[1] Wang Y, Skerry-Ryan R J, Stanton D, et al. Tacotron: Towards End-to-End Speech Synthesis[J]. Proc. Interspeech 2017, 2017: 4006-4010.*
> >
> > *[2] Shen J, Pang R, Weiss R J, et al. Natural tts synthesis by conditioning wavenet on mel spectrogram predictions[C]//2018 IEEE International Conference on Acoustics, Speech and Signal Processing (ICASSP). IEEE, 2018: 4779-4783.*
> >
> > *[3] Chen M, Tan X, Ren Y, et al. MultiSpeech: Multi-Speaker Text to Speech with Transformer[J]. Proc. Interspeech 2020, 2020: 4024-4028.*
> >
> > *[4] W. Ping, K. Peng, A. Gibiansky, S. O. Arik, A. Kannan, S. Narang, and J. Miller, “Deep Voice 3: Scaling Text-to-Speech with Convolutional Sequence Learning,” in International Conference on Learning Representations - ICLR, vol. 79, 2018, pp. 1094–1099.*

---

### Decision · Program_Chairs · 2021-09-27

**Decision:**

Accept (Poster)

**Comment:**

The paper proposes to apply Neural Transducers (popular for end-to-end ASR) to the problem of TTS, as well as TTS+ASR, which is agreed to be novel enough for NeurIPS by reviewers. The paper in its current form suffers however from its presentation, which raised many questions (mostly addressed by the rebuttal, though). In addition, it was noted that the experimental work could have been more convincing, as currently the quantitive results are no better than the baseline. Overall, the paper is thus very borderline, trending accept assuming the authors will heavily work on improving the presentation.